# Fungal Community Succession and Volatile Compound Changes during Fermentation of Laobaigan *Baijiu* from Chinese Hengshui Region

**DOI:** 10.3390/foods13040569

**Published:** 2024-02-14

**Authors:** Xuelian Yang, Jintao Yao, Ying Hu, Zichun Qin, Jingchao Li

**Affiliations:** Beijing Engineering and Technology Research Center of Food Additives, Beijing Technology and Business University (BTBU), Beijing 100048, China; m18600593866@163.com (J.Y.); m18618260251@163.com (Y.H.); qinzc08@163.com (Z.Q.); m18301396578@163.com (J.L.)

**Keywords:** Laobaigan *Baijiu*, core fungal community, volatile flavor, high-throughput sequencing, partial least square

## Abstract

To investigate the core fungal community succession and its effects of volatile compound production during different stages (D-1, D-2, D-3, E-4, E-5, and E-6) of Hengshui Laobaigan *Baijiu*, high-throughput sequencing (HTS) was carried out, accompanied by the identification and quantification of the volatile flavor compounds using headspace solid-phase coupled with gas chromatography–mass spectrometry (HS-SPME-GC-MS). HTS results demonstrated that the fungal community of stage D-1 was similar to that of E-4 after adding *Daqu*, while the richness and diversity of the fungal community were most prominent at stage E-6. Moreover, the addition of *Daqu* at the beginning of *Ercha* fermentation resulted in a significant increase in the relative abundances of the fungal community at the genus level, setting the stage for the production of volatile compounds. GC–MS analysis revealed the presence of a total of 45 volatile compounds. Combining the GC-MS result with the heat map and principal component analysis (PCA), the concentrations of volatile compounds were highest in stage E-5. Meanwhile, concentrations of esters, especially ethyl acetate, ethyl lactate, isoamyl acetate and ethyl hexanoate, were high in both stages E-5 and E-6. This indicated that stage E-5 was crucial to the fermentation process of Laobaigan *Baijiu*. Three fungal genera (Saccharomyces, Candida, and Pichia) were indicated as the core microbiota for the production of the main volatile flavor compounds of Laobaigan *Baijiu* through partial least square (PLS) analysis. The information provided in this study offered valuable insights into the fermentation mechanism of Laobaigan *Baijiu*, thereby serving as a theoretical framework for enhancing the quality of *Baijiu* and realizing cost-effective production.

## 1. Introduction

*Baijiu* (Chinese liquor) is classified into light aroma, strong aroma, rice aroma, et al., according to style characteristics [1]. Hengshui Laobaigan *Baijiu* is a representative brand of Laobaigan aroma, namely, the eleventh aromatic flavor type in China. It is popular for its transparency, mellow and luscious taste, and persistent aftertaste [2]. This specific distilled *Baijiu* is commonly made from pure sorghum and spontaneously fermented using original fermenters (medium-temperature *Daqu*). *Daqu* refers to a substantial, compacted mass that is produced by crushing sorghum alongside various grains, including barley and wheat. This process involves the admixture of water and subsequent compression, resulting in the formation of a lump enriched with polymicrobial enzymes. The fermentation process consists of two solid fermentation stages called *Dacha* and *Ercha*. Briefly, the first stage is to use sorghum as raw material and mix it with *Dipei* (grains that have undergone a process of steaming and fermentation), then distill to obtain the *Dacha Baijiu*, which is called *Dacha* fermentation. The second stage is to use the substrate of the *Dacha* fermentation as raw material, add *Daqu* once again to continue the fermentation process, and finally distill to obtain production, called *Ercha* fermentation [3].

Different aromatic flavor types are reflective of the composition of volatile compounds. Laobaigan *Baijiu* is characterized by its abundant content of ethyl acetate and ethyl lactate, with the ratio of these two volatile compounds ranging from 1:1.5 to 2, which create its unique styles [4]. The dynamic changes in microorganisms in different fermentation stages have significant implications for the composition of volatile compounds, which ultimately affects the flavor and quality of *Baijiu*. The spontaneous fermentation process of *Baijiu* facilitates the proliferation of diverse microorganisms that originate from *Daqu* or the environment. These microorganisms not only participate in the production of the main components of *Baijiu,* such as water and alcohols, but also produce part of the flavor components or their precursors, which is a great help in the formation of flavor components [5]. Among them, fungus is crucial to *Baijiu* fermentation, significantly affecting the final product’s aroma, texture, and acidity [6]. Therefore, it is imperative to delve into the core fungal species responsible for the generation of volatile compounds during Laobaigan *Baijiu* fermentation. In recent years, research on Laobaigan *Baijiu* has mainly focused on the flavor components and microorganisms themselves, respectively [2,5]. However, correlations between fungal community succession and volatile compound changes during fermentation have rarely been investigated. Therefore, an investigation of the correlations between the two sides is of significance for enhancing the quality of Laobaigan *Baijiu*.

Traditional methods for microflora analysis have been widely used to reveal microbial diversity in fermented drinks. However, these methods are not comprehensive, ineffective, or time-consuming [7,8]. High-throughput sequencing (HTS) has been widely adopted for the comprehensive analysis of complex microbial communities due to its low cost and short cycle time, which can provide a more in-depth and accurate assessment of complex microbial communities in diverse fermented foods [9,10,11]. The fungal community succession of fermented *Baijiu* products was investigated by the HTS technique, but similar studies of Laobaigan *Baijiu* were rarely found.

In this study, the dynamic changes in fungal communities at different fermentation stages were revealed using the HTS technique, while the corresponding volatile flavor compounds were identified and quantified by HS-SPME-GC-MS. For a deeper grasp of the core fungal communities involved in fermentation and their contribution to the volatile flavor compounds of Hengshui Laobaigan *Baijiu*, the relationship between them was explored by PCA and PLS. The results will lay the theoretical foundation for improving the taste, flavor, and overall quality of *Baijiu* and guide production.

## 2. Materials and Methods

### 2.1. Sample Collection

All Laobaigan *Baijiu* samples were collected from Hengshui *Baijiu* Industry Co. (Hengshui city, Hebei province, China), which is a representative company producing Laobaigan *Baijiu*. The fermentation process conditions for Laobaigan *Baijiu* are as follows: The moisture content of the fermentation system is maintained within a range of 53% to 56%. Additionally, the starch content falls between 20% and 22%, while the dosage of *Daqu* is also maintained at 20% to 22%. Furthermore, the fermentation temperature varies depending on the season. Specifically, during the summer months, the fermentation temperature of Laobaigan *Baijiu* should be maintained below the natural ambient temperature. Conversely, during other seasons, the fermentation temperature is maintained at a range of 15 °C to 22 °C. *Baijiu* samples we chose in this study were made during the summer months. Laobaigan *Baijiu* were fermented in floor tanks that possess dimensions of 80 cm in height and 80 cm in diameter, with a capacity ranging from 180 to 240 L. Samples were withdrawn from the upper (approximately 10 cm depth from the tank surface), the middle, and the lower (approximately 10 cm depth from the tank bottom) of a ferment tank at six specific fermentation periods (15, 30, 45, 60, and 75 days). The samples collected on the same date were thoroughly mixed and designated as D-1, D-2, D-3, E-4, E-5, and E-6, respectively, where D and E represent the two specific fermentation periods called *Dacha* and *Ercha*, and 1, 2, 3, 4, 5, and 6 represent the critical sampling day during the Laobaigan *Baijiu* fermentation.

### 2.2. Sample Pretreatment and DNA Extraction

*Baijiu* samples (5 g) were combined with 0.1 mol/L phosphate-buffered saline (PBS) solution (30 mL) in sterilized 50 mL centrifuge tubes, and these tubes were centrifuged at 9000 rpm for 5 min at room temperature. Subsequently, the supernatant was discarded, and the precipitate was resuspended in a 0.1 mol/L PBS solution (5 mL) to collect microorganisms. This process was replicated three times to ensure biological reproducibility. In order to extract the macro genomic DNA, the Simga-Aldrich GenEluteTM soil DNA isolation kit (Merck, Darmstadt, Germany) was utilized, following the manufacturer’s instructions. The extracted DNA from the triplicate samples was pooled, and the quality and quantity of the mixed DNA were assessed via a NanoDrop ND2000 spectrophotometer (Thermo Fisher Scientific, Waltham, MA, USA). The DNA concentration was adjusted to 20 nanograms per microliter. The integrity of the extracted DNA was verified through agarose gel electrophoresis and stored at −80 °C until further analysis.

PCR amplification and sequencing were conducted utilizing the previously established methodology [12]. The ITS2 region of the fungal ITS DNA gene was amplified using the primers ITS1: 5′-TCCGTAGGTGAACCTGCGG-3′ and ITS4: 5′-TCCTCCGCTTATTGATATGC-3′. The PCR reaction mixtures contained: 1.0 μL of genomic DNA (20 ng/μL), 1.0 μL of each primer (10 μM), 0.3 μL of Ex Taq (5 U/μL), 2.5 μL of Ex Taq buffer (10×), 2.0 μL of dNTP mixture (2.5 mM), and 17.2 μL of double-distilled water. PCR amplification was carried out as follows: the initial denaturation step at 94 °C for 5 min, followed by 30 cycles of denaturation at 94 °C for 5 min, annealing at 55 °C for 30 s, extension at 72 °C for 90 s, and a final extension at 72 °C for 10 min. The resulting PCR products were purified from a 1% (*w*/*v*) agarose gel and quantified using a NanoDrop ND2000 spectrophotometer (Thermo Fisher Scientific, Waltham, MA, USA).

### 2.3. Alpha Analysis

ITS DNA gene sequencing libraries were used for constructing high-throughput sequencing. The purified PCR products were sequenced on the Illumina MiSeq platform (Illumina, San Diego, CA, USA) at Personal Biotechnology Co., Ltd. (Shanghai, China) [13]. The double-end sequences that were primed by mass were paired according to overlapping bases using FLASH software (version 34.0.0.277) to obtain the effective sequence of each sample [14]. Raw sequencing reads were quality-filtered and analyzed using QIIME (version 1.7.0). Ambiguous and chimera sequences were eliminated using the method of UCHIME in MOTHUR (version 1.31.2). High-quality sequences were clustered into operational taxonomic units (OTUs), defined by a sequence similarity threshold of 97%, using the UCLUST method embedded in QIIME (version 1.7.0) [15]. The sequence with the highest abundance in each OTU was designated as the representative sequence. These representative OTUs were then classified into various taxonomic levels (phylum, class, order, family, and genus) via the Ribosomal Database Project (RPD) [16]. Alpha diversity indices, including Chao richness indices, observed species (Sobs), Shannon, and Simpson, were calculated using the summary single command in MOTHUR to analyze species diversity and richness. The corresponding rarefaction curves were drawn using QIIME (version 1.7.0).

### 2.4. Beta Analysis

Beta diversity was facilitated using the QIIME software suite (version 1.7.0). This software computed the differences between the groups of samples using the weighted UniFrac distance [17]. Prior to performing clustering analysis, principal coordinate analysis (PCoA) was employed to reduce the dimensionality of the original dataset in the FactoMineR package. The resulting visualizations were then generated by making use of the ggplot2 package in R software (version 2.15.3). This approach enabled the visualization of complex multidimensional data through simplified dimensional reduction techniques based on PCoA. A hierarchical clustering tree was constructed to illustrate the similarities and differences in the constituent structures of various sample species, which were inferred by QIIME software (version 1.7.0) and visualized by R software (version 2.15.3). To further quantify the diversity of species abundance distributions across the samples, significant differences were measured using a sample distance heatmap generated by R software (version 2.15.3) along with the Fast UniFrac algorithms and FastTree. The unweighted pair group method and the arithmetic mean (UPGMA) approach, a type of hierarchical clustering, were used to measure the distance matrix using the average-linkage method. The UPGMA approach was implemented through the QIIME software (version 1.7.0) [18].

### 2.5. Volatile Profiles Analyzed by Headspace SPME Combined with GC-MS

Volatile compounds within fermented *Baijiu* were analyzed using HS-SPME-GC-MS. Prior to initial use, the SPME extraction head was aged under the specific condition of 280 °C for 40 min at the injection port of the gas chromatograph. To summarize, a 7.98 mL treated sample was combined with 2 g NaCl in a 20 mL headspace flask. Subsequently, 20 microliters of the internal standard (4-methyl-2-pentanol) was added. The headspace flask was incubated in a 50 °C water bath for 20 min. Volatile compounds were then collected using a 50 μm PDMS/CAR/DVB fiber provided by Supelco (Bellefonte, PA) for a duration of 45 min. After extraction, the fiber was immediately inserted into the injection port of an TSQ 8000 Evo GC–MS (Thermo Fisher Scientific, Waltham, MA, America) for the desorption step at 280 °C for 2–5 min.

The analytical conditions were as follows: Chromatographic separation was completed on a DB-WAX capillary column (30.0 m × 0.25 mm × 0.25 μm, Agilent, City of Santa Clara, CA, USA). Nitrogen was used as a carrier gas at a constant flow rate of 1 mL/min in splitless mode, and the injector temperature was 250 °C. The oven temperature was maintained at 35 °C for 2 min, then heated to 100 °C at 2 °C/min and held for 5 min, and then heated to 230 °C at 10 °C/min and held for 2 min. Finally, increase to 280 °C at 10 °C/min and hold for 3 min. The interface temperature was maintained at a consistent 230 °C during the analytical process. Mass spectra were acquired in the electron impact (EI) mode, utilizing an ionization energy of 70 eV and operating in the full scan range of 30 to 350 atomic mass units (amu). The tentative identification of compounds was facilitated through a meticulous comparison of the obtained mass spectra with the reference data from the NIST 1.1 mass spectral database. Quantitative determinations were conducted with precision, using 4-Methyl-2-pentanol as an internal standard at a volume of 20 μL and a final concentration of 51.15 μg/L. Integrating the selected 4-Methyl-2-pentanol to gain the related peak areas of volatile compounds, the quantification was calculated accurately.

### 2.6. Statistical Analysis

The changes in volatile compounds and classification by cluster analysis were visualized and analyzed via heat maps using TBtools-II software. ANOVA was employed to detect statistically significant differences (*p* < 0.05) among the mean values. PLS analysis was conducted using Simca-14.1 software (Sartorius Stedim, Gottingen, Sweden) to reveal the potential relationship between the microorganisms and volatile flavors. PCA analysis using Simca-14.1 software (Sartorius Stedim, Gottingen, Sweden). Each dataset was performed three times in parallel.

## 3. Results and Discussion

### 3.1. Richness and Diversity of Fungal Communities in Different Stages of Fermentation

After sequencing the samples, rigorous quality control measures were implemented, including the removal of low-quality reads and chimera filtering, to obtain valid data. A total of 822,969 high-quality sequencing reads of fungal ITS DNA genes were generated from the samples (220–500 bp). On average, over 27,432 reads per sample were achieved. Furthermore, the rarefaction curves for all samples exhibited a distinct saturation plateau, with coverage exceeding 99%. This revealed that the vast majority of fungal phylotypes present in the Baijiu samples have been successfully identified.

Chao1 and observed species (Sobs) values were calculated to determine species richness for Laobaigan *Baijiu* samples in different fermentation stages, and Shannon and Simpson indices were calculated to determine fungal diversity [19]. As shown in Table 1, combining Chao1 values with Sobs values of the whole fermentation process, the species richness of the fungal community was ranked as follows: E-6 > D-3 > D-1 > E-4 > E-5 > D-2. In regard to fungal diversity, the Shannon and Simpon indices of the fungal community were both ranked as follows: E-6 > D-1 > E-4 > D-3 > E-5 > D-2. The results showed that the richness and diversity of the fungal community in *Ercha* stages were higher than those in *Dacha*. The highest and lowest values of alpha diversity, such as observed species, Chao1 richness, and Shannon indices, were both found in stages E-6 and D-2, respectively, which indicated that the fungal richness and diversity were highest at stage E-6 while lowest at stage D-2. This showed that the introduction of microorganisms in stages D-1 and E-4 disrupted the balance of the original fungal community; some of the initial fungal species died out, while those more adapted to the fermentation environment survived. The microbial population reached a new equilibrium in stage D-2. The addition of *Daqu* in stage E-4 supplemented the fungal microorganisms so as to achieve the purpose of making full use of the substrate.

Beta diversity was utilized to compare the compositions of microbial communities as well as assess differences between them. The fundamental result of this comparative analysis was a distance matrix, which delineated the community-level variations between two different samples. The community structures were analyzed using average linkage clustering [18]. At the genus level, the microbiota in stages D-1 and E-4 exhibited significant differences from those of stages D-2, D-3, E-5, and E-6, as evident from the hierarchical clustering tree constructed based on weighted UniFrac distances (Figure 1). In detail, the fungal community of stage D-1 differed significantly from that of stages D-2 and D-3, while the fungal community of stage D-2 was similar to that of stage D-3. In addition, the fungal community of stage E-6 had a significant difference from other samples. This indicated that the fungal community changed dramatically at stages D-1 and E-4 and that the microbial structure was gradually established after stages D-2 and E-5.

To further demonstrate the differences in species diversity between samples, a PCoA analysis of fungal structure during Laobaigan *Baijiu* fermentation was used to display the magnitude of differences between individual samples (Figure 2). The distance of different samples reflects the closeness of sample composition [20]. With the contribution of 79.52% from the first principal coordinate (Pco1), the fungal composition of stage D-1 was close to that of stage E-4, and the fungal composition of stages D-2, D-3, and E-5 was close to that of stage E-6. With the contribution of 8.61% from the second principal coordinate (PCo2), the fungal composition of stages D-1, D-2, D-3, and E-4 was close to that of stage E-5. In addition, the five points of E-6 were scattered, and the distance between them was large in both the first and second principal coordinates, indicating that there were obvious differences within the group of E-6 samples. This was due to the fact that environmental microorganisms introduced by operations such as *Baijiu* extraction at the end stage of fermentation affected the fungal composition of Laobaigan *Baijiu*. A combined analysis of the contributions of the first and second principal coordinates showed that the fungal composition of D-1 was similar to that of E-4, while the richness and diversity of the fungal community were most prominent at stage E-6. This illustrated that the fungal composition of Hengshui Laobaigan *Baijiu* samples showed good reproducibility between the two fermentation stages. The addition of *Daqu* enriched the fungal microorganisms in *Baijiu* samples, providing abundant microorganisms for *Ercha* fermentation, and the adaptation of the fungal community to the fermentation environment led to the cyclicity of the fungal community succession.

### 3.2. Relative Abundance of Fungal Communities

To investigate the dynamic changes in the fungal community during the fermentation of Laobaigan *Baijiu*, effective sequencing reads were classified into different taxon levels by an RDP classifier. The distribution of the OTUs was categorized at the phylum and genus levels (Figure 3). In detail, there were four phyla detected in Laobaigan *Baijiu*. Ascomycota (90.57%) accounted for the highest proportion, followed by Zygomycota (7.12%), Basidiomycota (1.65%), and unidentified fungi (0.025%). Ascomycota was identified as the crucial fungal phylum throughout the whole *Baijiu* fermentation process, which agreed with the previous results for Laobaigan *Baijiu* [4]. Ascomycota showed an increasing trend at stage D-2 but experienced a small decline at stage D-3 (Figure 3a). In contrast, Zygomycota decreased dramatically at stage D-2 and then increased slightly at stage D-3. Similar changes occurred during the *Ercha* fermentation stages.

At the genus level, the relative abundances of the top twenty fungal OTUs were calculated (microorganisms with a relative abundance of less than 0.07% were grouped together under the umbrella term “others”). As shown in Figure 3b, Saccharomyces, Saccharomycetales_unidentified_1, and Candida were identified as the crucial fungal genera throughout the whole Laobaigan *Baijiu* fermentation process. Saccharomyces accounted for the highest proportion (50.80%), followed by Saccharomycetales_unidentified_1 (19.3%), Candida (11.94%), Mucor (4.88%), Aspergillus (3.04%), and Rhizopus (1.97%). In stage D-1, Saccharomycetales_unidentified_1, Candida, and Mucor were the initially dominant genera. Mucor, which mainly grew in the lower temperature period of Laobaigan *Baijiu* fermentation, was mostly harmful microorganisms, but small amounts of Mucor favored protein breakdown. Candida possessed a robust capacity for ester compound production, coupled with a certain alcoholic fermentation ability, thereby establishing the groundwork for imparting unique flavor profiles to Laobaigan *Baijiu*. These genera exhibited optimal growth and proliferation at lower fermentation temperatures. This observation was in line with the reported higher presence of Mucor and Candida at stage D-1 [21]. The relative abundances of these three genera decreased rapidly in stage D-2, while the relative abundance of Saccharomyces increased rapidly up to a maximum (94.13%) and became the dominant genera. Alcohol compounds were mainly generated in the early stages of *Baijiu* brewing, and Saccharomyces had a strong alcohol production capacity. This accounted for the significant increase in Saccharomyces at stage D-2. As fermentation processed, these alcohol compounds were gradually converted into esters and other compounds under suitable conditions [4]. In stage D-3, the relative abundances of Saccharomyces and Saccharomycetales_unidentified_1 decreased gradually to 75.33% and 1.60%, respectively. In contrast, the relative abundance of Candida experienced a dramatic upward trend from stage D-2 to D-3, in which the value increased from 0.68% to 12.30%. At this stage, Candida used organic acids and inorganic salts in raw materials to produce esterases. These enzymes catalyzed the esterification reaction between organic acids and alcohols, resulting in the formation of the corresponding esters [22]. The values of Aspergillus, Rhizopus, Pichia, Pencilliumn, Wickerhamomyces, et al. showed a tendency to initially decrease and then increase during the *Dacha* fermentation. Aspergillus had strong saccharification ability as well as generating a variety of enzymes, with a great catalytic ability for macromolecule degradation such as starch and proteins, which has positively contributed to *Baijiu* quality. Studies have also shown that Aspergillus can also inhibit the growth of harmful microorganisms by producing acidic proteases to lower pH, which also ensures the relative safety of *Baijiu* [23]. Interestingly, the relative abundance of the Trametes genus remained zero in stages D-1 and D-2; the highest value appeared in stage D-3 (less than 1%). The relative abundances of the remaining fungal, including Monascus, Debaryomyces, Davidiella, and Mucorales_unidentified_1 did not change greatly during the fermentation process. The results indicated that the original balance of the fungal community was disturbed after *Daqu* was added to the tank. The original main genera, such as Saccharomycetales_unidentified_1, Candida, and Mucor, were not as well adapted to the environment as Saccharomyces and required a certain period of time for domestication before they could take over the dominance of the fungal community.

In the *Ercha* fermentation, the changes in the relative abundances showed good reproducibility. Notably, the addition of *Daqu* resulted in a rapid increase in the relative abundances of Saccharomycetales_unidentified_1, Candida, and Aspergillus. Although Saccharomyces also displayed the same change as in the *Dacha* fermentation, its relative abundance was not as large as that of the *Dacha* fermentation. In contrast, Saccharomycetales_unidentified_1, Candida, and Aspergillus demonstrated a strong reproductive capacity. This indicated that these three fungal introduced by *Daqu* could better utilize the substrate after the completion of the *Dacha* fermentation. The addition of *Daqu* at the beginning of *Ercha* fermentation resulted in a significant increase in the relative abundance of the fungal community at the genus level. At last, fungal communities came to a new balance at stage E-6, where Saccharomyces, Saccharomycetales_unidentified_1, Candida, Pichia, Trametes, Rhizopus, and Aspergillus also occupied larger shares, setting the stage for the production of volatile compounds.

### 3.3. Volatile Compounds Identified Using GC-MS

Volatile compounds in different fermentation stages of Laobaigan *Baijiu* were identified by HP-SPME-GC-MS. As Table 2 reveals, 45 volatile compounds were identified totally in the samples, which consist of 10 alcohols, 17 esters, 7 aldehydes, 4 ketones, 3 acids, 1 furan, 1 aromatic, 1 lactone, and 1 sulfur-containing compound. Among them, ethyl acetate, ethyl lactate, and ethyl isovalerate, defined as the typical core volatile compounds of Laobaigan *Baijiu*, have also been found in other aromatic flavors of *Baijiu* [24].

During the fermentation of Laobaigan *Baijiu*, the overall quantity of volatile compounds exhibited a gradual increase over time, with specific categories of volatile components undergoing alterations at distinct stages. The main substances in Laobaigan *Baijiu* were alcohols (>10), with a mass fraction of 66.78% of the total volatile compounds. Among them, ethanol, 3-methylbutanol, 1-hexanol, and 1-propanol were the main alcohol compounds. The content of ethanol and 3-methylbutanol increased gradually and reached a peak at stage E-6. This was associated with Saccharomyces always being the main dominant genera. As fermentation proceeded, the content of ethanol was basically stabilized due to inhibition from other metabolites and consumption involved in other physiological metabolic activities, such as esterification [25].

Esters were the most important volatile compounds, which had a great influence on the quality of Laobaigan *Baijiu*, and a total of 17 ester compounds were quantified. The content of these esters accumulated dramatically at stage D-3 and increased gradually up to the highest (42.452 μg/L) at stage E-6. This was associated with the relative abundances of Candida, Aspergillus, Pichia, and other fungal increasing dramatically at stage E-3. There was a small decrease at stage E-4 due to the fact that the addition of *Daqu* at the beginning of *Ercha* fermentation had an effect on the composition of the fermentation substrate. Candida and Pichia have been reported to have efficient ester production abilities. A large quantity of ester compounds was produced via condensation reaction in the presence of esterase [26]. Usually, the ester compounds have the characteristics of a fruity aroma. It is the most important volatile substance in *Baijiu*, which determines its aromatic flavor type and quality [27]. Among them, ethyl acetate, isoamyl acetate, ethyl hexanoate, and ethyl lactate were the main volatile compounds. Ethyl acetate has the characteristics of a fruity aroma and tastes a little astringent, while ethyl lactate has the characteristics of a grassy aroma and tastes a little sweet. These two ester compounds determine the front-scent and after-scent of Laobaigan *Baijiu*, respectively. Ethyl hexanoate has the characteristics of an apple aroma, which makes the unique complex aroma of ethyl acetate and ethyl lactate more evident. Isoamyl acetate has the characteristics of a pear and apple aroma [28]. These ester compounds interact with each other to form the unique aroma of Laobaigan *Baijiu*.

Compared with alcohols and esters, the content of aldehydes was relatively less abundant. A total of six aldehyde compounds were detected in the fermentation of Laobaigan *Baijiu,* and 1,1-diethoxyethane, acrolein, and hexanal were the main aldehyde compounds. The content of aldehydes increased at stage D-2 and then experienced a decrease at stage D-3. Similar changes occurred during the *Ercha* fermentation stage. In terms of changes in concentration, aldehydes were mainly produced in stages D-2 and E-5, along with alcoholic fermentation, and they rapidly enriched as the yeast multiplied. Afterwards, the aldehydes show a decreasing trend due to changes in the physicochemical properties of the fermentation system and the utilization of the aldehydes to produce organic acids [29]. Aldehydes contributed significantly to flavor. Usually, aldehydes have an obvious aroma-boosting effect on Laobaigan *Baijiu* and make it mellow and pure. 1,1-Diethoxyethane is the highest aldehyde compound with a strong volatility and irritation, but with the appropriate ratio of acetaldehyde, it can significantly enhance the aroma of Laobaigan *Baijiu* and eliminate the negative impact of ethyl lactate on the taste.

Other volatile compounds, such as ketones, acids, and furan compounds, although relatively low in content, also have a significant impact on the aroma composition. 2-Pentanone has the characteristics of a sweet, fruity aroma and an ethereal wine aroma. 2-Heptanone has the characteristics of a nutty aroma. Acetic acid has the characteristics of a rice vinegar aroma and harmonizes the taste [3]. 2-Butylfuran has the characteristics of a fruity aroma and plays an important role in Laobaigan *Baijiu* due to its low odor thresholds [30]. All of the volatile compounds mentioned above contributed to the unique aroma of Laobaigan *Baijiu*.

Furthermore, the GC-MS results presented in Table 2 were subjected to heat map visualization and principal component analysis (PCA) for comprehensive interpretation, as shown in Figure 4 and Figure 5, respectively. As evident from the illustration provided in Figure 4, the color gradient from red to blue signifies the varying abundance of compounds, ranging from higher to lower levels. The concentrations of ethanol, 3-methylbutanol, ethyl acetate, ethyl lactate, isoamyl acetate, ethyl hexanoate, acetic acid, hexanoic acid, octanoic acid, and 2-Butylfuran increased to their maximum in stage D-6. The highest values of octanal, acrolein, 2-pentanone, hexanal, 2-heptanone, furfural, acetophenone, 2,3-butanedione, 2-methylpropionaldehyde, 1,1-diethoxyethane, and acetaldehyde appeared in stage E-5, but the final concentrations of these volatile compounds decreased dramatically in stage E-6 (Figure 4). These volatile compounds played crucial roles in the formation of the aromatic flavor type of Laobaijian *Baijiu* due to their unique flavors. Therefore, combining the types with the contents of various volatile compounds, stage E-5 was critical to the formation of the unique aroma of Laobaijian *Baijiu*, and the fermentation period could be brought forward to end at stage E-5.

Figure 5a shows that the contribution rate of the first principal component was 76.6%, the second principal component was 20.4%, and the cumulative contribution rate was 97.0%, indicating that the PCA analysis captured the majority of the information contained within the original variables and that the results of the PCA analysis had strong reliability. Based on the score plot (Figure 5a), the samples of stages D-3, E-4, and E-6 were distributed closer, indicating that the volatile composition of these three fermentation stages had a high similarity. In contrast, the position of sample D-1 was far from the position of the points for other fermentation periods of samples, which demonstrated the significant difference in the composition of the volatile components.

As shown in Figure 5b, the position of the most volatile compounds exhibited a strong correlation with the location of the E-5 and E-6 samples on the score plot presented in Figure 5a. This observation suggested a significant positive association between the samples of E-5 and E-6 and the most volatile compounds. Moreover, the locations of D-3 and E-4 in Figure 5a also appeared to exhibit a certain degree of correlation with the most volatile compounds in Figure 5b, indicating that ethyl ether extraction also had a certain positive correlation. Which is to say, the four fermentation stages mentioned above played a good role in the composition of the most volatile compounds. In contrast, D-1 and D-2 showed poor correlation with the most volatile compounds because they were far away from the most volatile compounds on the loading plot (Figure 5b). The aforementioned findings were in concordance with the previously conducted heat map visualization analysis and GC-MS analysis results.

### 3.4. PLS-Based Correlation Analysis between Fungal and Volatile Compounds

To further investigate whether the generation of certain volatile compounds was mediated by one or more fungal communities, the PLS method was used to analyze the correlation between fungal community succession and volatile compound changes during fermentation. The fungal genera and volatile compounds were defined as variables X and Y in the PLS model, respectively. There were 10 variables (X) and 45 variables (Y) analyzed.

The two large circles in the center and the outermost part of the Figure 6 represented the 50% and 100% confidence intervals, respectively, and variables located between the two circles indicated that they could be well explained by the model. It could be seen from Figure 6 that all of the volatile compounds were located within the 50% to 100% confidence interval, indicating that the model had good explanatory power between the variables. Candida had the highest correlation with 8 of the main volatile compounds and had positive correlation coefficients with ethyl isobutyrate (13), ethyl butanoate (14), ethyl isovalerate (15), isoamyl acetate (16), ethyl valerate (17), ethyl hexanoate (18), ethyl lactate (19), and ethyl heptanoate (20). Pichia was well correlated with 5 of the main volatile compounds and had positive correlation coefficients with ethanol (1), 3-methylbutanol (4), ethyl phenylacetate (26), acrolein (30), and hexanal (31). Trametes was correlated well with 5 of the main volatile compounds and had a positive correlation with ethanol (1), 3-methylbutanol (4), acrolein (30), acetophenone (37), and octanoic acid (40); Saccharomyces genera correlated strongly with pentanol (5) and ethyl caprylate (21). Wickerhamomyces was better correlated with 2,3-butanedione (34). *Mucor* was better correlated with ethyl tetradecanoate (27), 2-methylpropionaldehyde (29), 2,3-butanedione (34), and furfural (42). Aspergillus and Penicillium were better correlated with dimethyl trisulfide (45). Rhizopous was strongly correlated with acetophenone (37) and octanoic acid (40).

It showed that the position of the most volatile compounds was well correlated with the position of Saccharomyces, Pichia, Candida, Trametes, and Rhizopous. It is the most relevant for interpreting the formation of volatile compounds. Especially for Pichia and Candida, the position of the most volatile compounds distributed around them indicated that these two genera were the two most important contributors to the formation of flavors during *Baijiu* fermentation. In addition, Aspergillus, Penicillium, Wickerhamomyces, and Mucor showed a good correlation with certain volatile compounds but showed a poor correlation with the most volatile compounds. The position of Torulaspora was far from all of the volatile compounds on the loading plot (Figure 5b), which indicated that Torulaspora is not the core genera that produces volatile compounds, which was in line with the previous study [4].

## 4. Conclusions

This study revealed the dynamic changes in fungal communities and volatile flavor during the fermentation of Hengshui Laobaigan *Baijiu* for the first time. Species classification and abundance analysis showed that Saccharomyces, Saccharomycetales_unidentified_1, Candida, Mucor, Aspergillus, and Rhizopus were the predominant genera during the fermentation process. Furthermore, a total of 45 volatile compounds were detected in Laobaigan *Baijiu* samples via HS-SPME-GC-MS, including alcohols, esters, aldehydes, ketones, and others. The heat map and PCA showed that stage E-5 was critical to the formation of the unique aroma of Laobaijian *Baijiu,* and the fermentation period could be brought forward to end at stage E-5. The PLS-based correlation analysis showed that the fungal Saccharomyces, Candida, Pichia, and Trametes had important roles in the formation of volatile compounds. The above results indicated that the genera Saccharomyces and Candida played a pivotal role in Laobaigan *Baijiu* fermentation. Specifically, Candida, endowed with esterification capabilities, was primarily responsible for the biosynthesis of ester compounds. As the core volatile compounds of *Baijiu*, the content and proportion of esters, especially the four major ethyl esters (ethyl acetate, ethyl lactate, isoamyl acetate, and ethyl hexanoate), largely determined the quality and taste of *Baijiu*. In actual production, the logarithmic period of the growth of Candida and other fungal related to ester production should be prolonged by controlling the fermentation process (fermentation temperature, pH, moisture content, etc.) in order to promote their massive growth and reproduction, thus increasing the concentration of esters such as ethyl acetate, ethyl lactate, isoamyl acetate and ethyl hexanoate. Moreover, considering the cost and the quality of Laobaigan *Baijiu*, the fermentation period could be brought forward to end at stage E-5. This study provides a good entry point for an in-depth understanding of the fermentation mechanism of *Baijiu* and provides a theoretical basis for the improvement in the quality, flavor, and industrial production of *Baijiu*.

## Figures and Tables

**Figure 1 foods-13-00569-f001:**
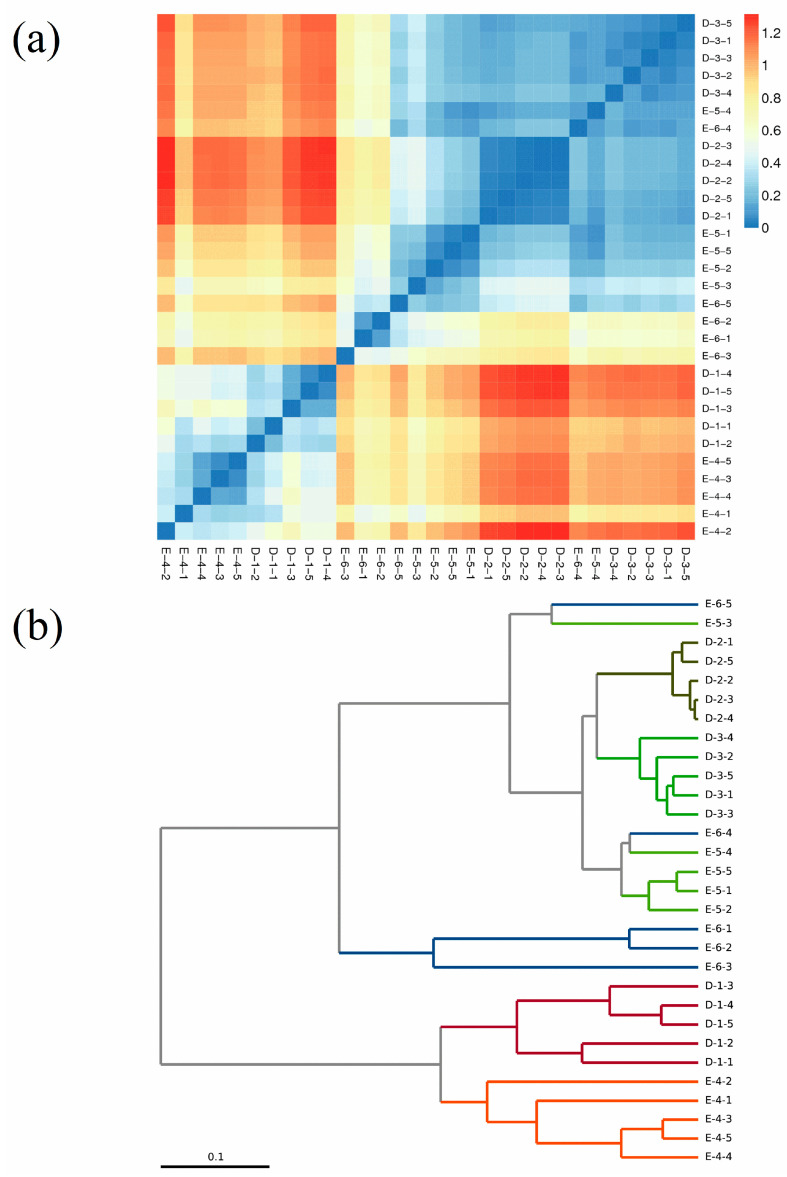
Samples distance heatmap and hierarchical clustering tree on genus level: (**a**) sample distance heatmap on genus level; (**b**) hierarchical clustering tree on genus level. D-1 (
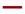
), D-2 (
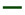
), D-3 (
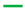
), E-4 (
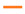
), E-5 (
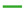
) and E-6 (
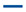
) represent samples from six fermentation stages.

**Figure 2 foods-13-00569-f002:**
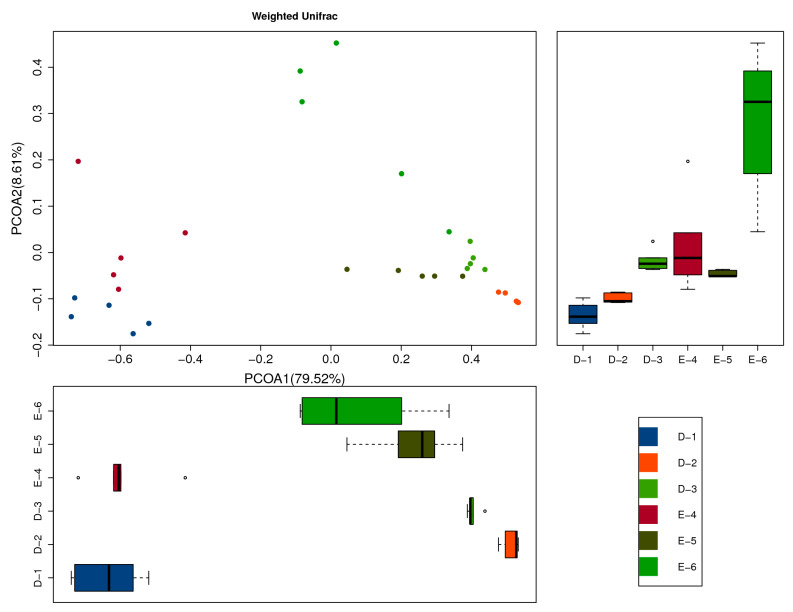
Principal coordinates analysis of fungal formation during the fermentation based on the weighted UniFrac algorithm. The dots in the figure represent individual samples, with different colors indicating that the samples belong to different subgroups.

**Figure 3 foods-13-00569-f003:**
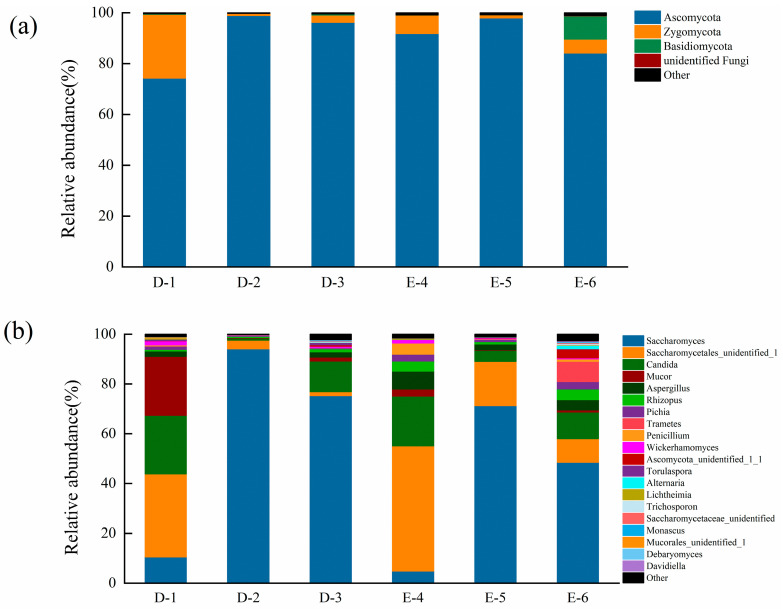
Taxonomic classification of sequences from fungal community of Laobaigan *Baijiu* samples. D-1-E-6 represent the *Baijiu* samples collected at six specific fermentation stages. (**a**,**b**) represent the classification at the phylum and genus level, respectively.

**Figure 4 foods-13-00569-f004:**
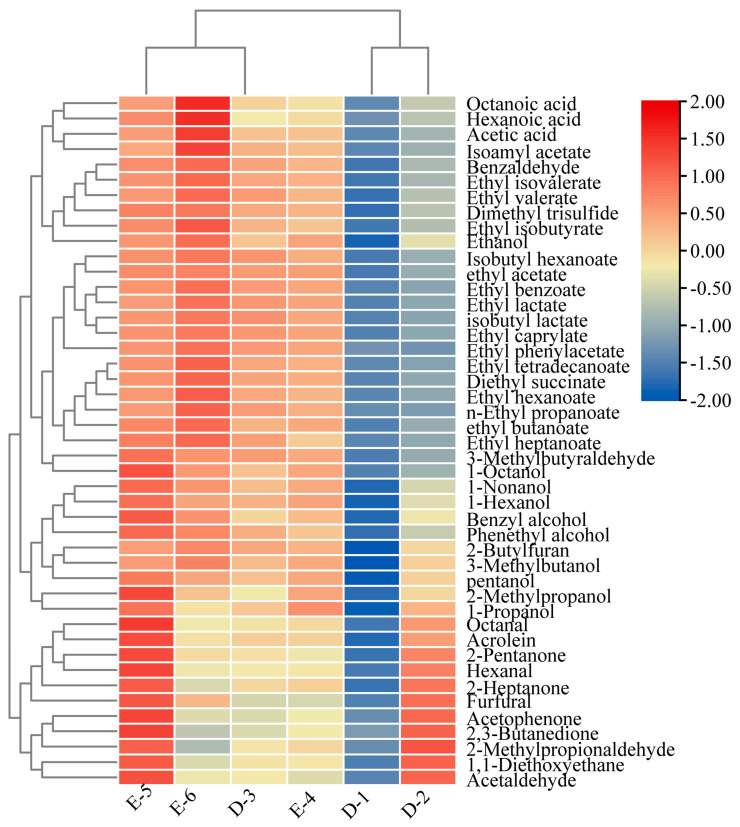
Heat map of volatile compounds identified in Laobaigan *Baijiu*.

**Figure 5 foods-13-00569-f005:**
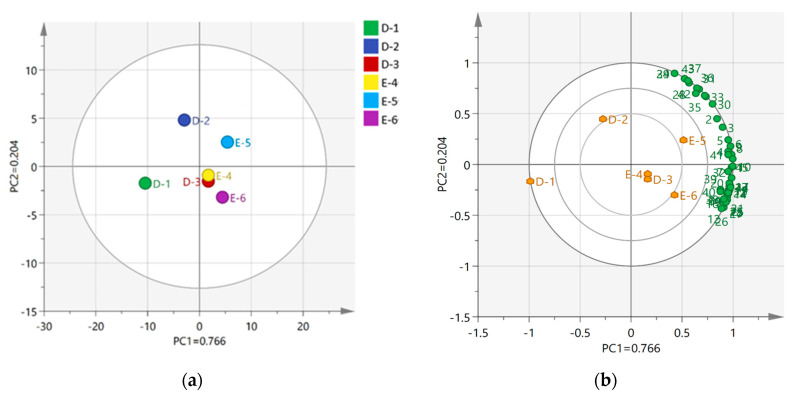
(**a**) Score plot of volatile compounds in Laobaigan *Baijiu* using HS-SPEM-GC−MS analysis; (**b**) Loading plot of volatile compounds in Laobaigan *Baijiu* using HS-SPEM-GC−MS during the PCA analysis.

**Figure 6 foods-13-00569-f006:**
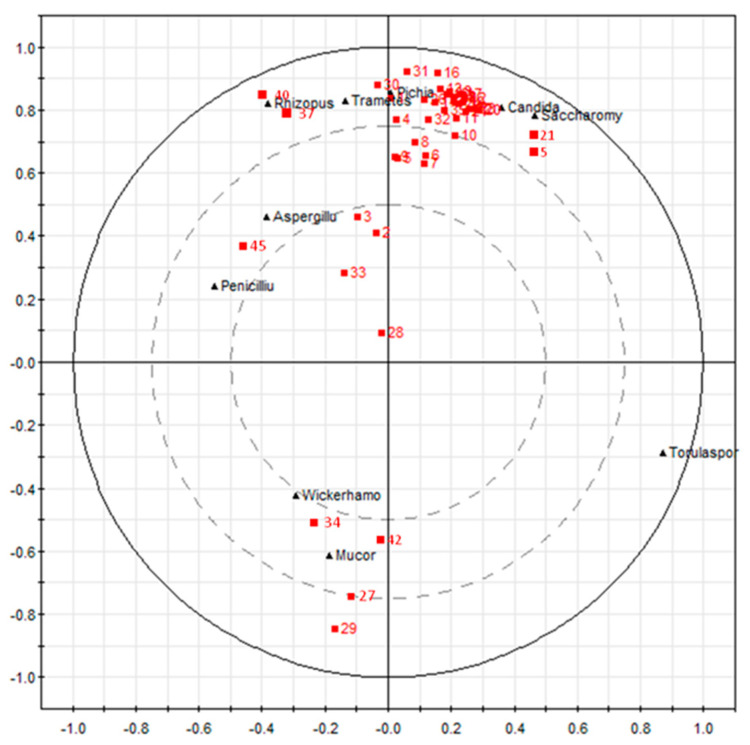
Correlation analysis between the fungal community and volatile compounds during Laobaigan *Baijiu* fermentation by PLS model.

**Table 1 foods-13-00569-t001:** Sequencing data statistics for different Laobaigan *Baijiu*.

Sample	Clean Reads	OUT ^1^	Sobs ^2^	Chao1 ^2^	Shannon ^2^	Simpson ^2^	Coverage ^2^
D-1	35,333	101	79.4	112.77	2.6767	0.764459	99.3
D-2	35,795	63	41	56.791	0.8852	0.261388	99.7
D-3	37,418	161	109.6	158.01	2.0635	0.531775	99.4
E-4	36,706	98	78.8	100.02	2.5793	0.691885	99.6
E-5	34,280	96	75.2	97.76	1.8008	0.526424	99.5
E-6	36,873	208	131.4	156.03	3.1106	0.725503	99.4
Value *p*			0.000168	0.0004	0.000368	0.00083	0.000958

^1^ The operational taxonomic units (OTUs) were defined with 97% similarity level. ^2^ Chao1 = Sobs + n_1*_(n_1_ − 1)/2(n_2_ + 1), where n_1_ is the number of OTUs containing only one sequence and n_2_ is the number of OTUs containing two sequences; Shannon (H’ = −∑Pi ln (Pi), where Pi is the proportion of taxon i) and Simpson (D = 1 − ∑Pi2); coverage (G = 1 − n/N, where n is the number of phylotypes that have been sampled once and N is the total number of individuals in the sample).

**Table 2 foods-13-00569-t002:** Changes in the volatile compounds during Hengshui Laobaigan *Baijiu* fermentation.

No.	Compounds	Concentration (μg/L) ^2^					
		D-1	D-2	D-3	E-4	E-5	E-6
	Alcohols (10)						
1	Ethanol	3.341 ± 0.045 ^a^	29.893 ± 0.104 ^a^	38.780 ± 0.034 ^a^	44.657 ± 0.141 ^a^	47.785 ± 0.036 ^a^	53.451 ± 0.007 ^a^
2	1-Propanol	ND ^1^	1.023 ± 0.031 ^b^	0.934 ± 0.003 ^d^	1.178 ± 0.002 ^a^	1.309 ± 0.004 ^b^	0.824 ± 0.003 ^a^
3	2-Methylpropanol	ND	0.102 ± 0.009 ^d^	0.092 ± 0.006 ^d^	0.131 ± 0.002 ^d^	0.182 ± 0.002 ^a^	0.114 ± 0.001 ^a^
4	3-Methylbutanol	2.012 ± 0.014 ^a^	20.438 ± 0.122 ^a^	22.091 ± 0.051 ^b^	23.598 ± 0.084 ^c^	24.781 ± 0.006 ^d^	26.900 ± 0.002 ^a^
5	Pentanol	ND	0.0934 ± 0.011 ^d^	0.101 ± 0.004 ^d^	0.112 ± 0.009 ^d^	0.131 ± 0.003 ^d^	0.113 ± 0.001 ^d^
6	1-Hexanol	0.207 ± 0.011 ^b^	0.903 ± 0.016 ^a^	1.238 ± 0.013 ^a^	1.301 ± 0.003 ^d^	1.526 ± 0.003 ^a^	1.290 ± 0.005 ^a^
7	1-Octanol	ND	0.112 ± 0.073 ^d^	0.320 ± 0.005 ^d^	0.368 ± 0.005 ^b^	0.520 ± 0.003 ^a^	0.392 ± 0.002 ^a^
8	1-Nonanol	ND	0.089 ± 0.016 ^d^	0.134 ± 0.002 ^d^	0.147 ± 0.006 ^d^	0.187 ± 0.000 ^d^	0.161 ± 0.002 ^c^
9	Benzyl alcohol	ND	0.045 ± 0.003 ^c^	0.053 ± 0.004 ^d^	0.060 ± 0.004 ^d^	0.087 ± 0.003 ^c^	0.071 ± 0.000 ^d^
10	Phenethyl alcohol	ND	0.023 ± 0.006 ^d^	0.043 ± 0.002 ^d^	0.038 ± 0.008 ^d^	0.056 ± 0.002 ^d^	0.050 ± 0.002 ^d^
	Subtotal	5.560	52.721	63.786	71.590	76.564	83.366
	Esters (17)						
11	Ethyl acetate	0.023 ± 0.003 ^a^	3.901 ± 0.047 ^a^	13.784 ± 0.079 ^a^	13.672 ± 0.003 ^d^	14.778 ± 0.012 ^a^	15.321 ± 0.002 ^b^
12	n-Ethyl propanoate	ND	0.023 ± 0.004 ^b^	0.243 ± 0.007 ^a^	0.221 ± 0.007 ^d^	0.245 ± 0.003 ^d^	0.314 ± 0.001 ^b^
13	Ethyl isobutyrate	ND	0.093 ± 0.006 ^c^	0.210 ± 0.004 ^b^	0.191 ± 0.007 ^d^	0.249 ± 0.000 ^d^	0.302 ± 0.001 ^c^
14	Ethyl butanoate	ND	0.539 ± 0.002 ^a^	1.934 ± 0.009 ^a^	2.013 ± 0.004 ^c^	2.352 ± 0.004 ^a^	2.654 ± 0.006 ^a^
15	Ethyl isovalerate	ND	0.099 ± 0.015 ^d^	0.265 ± 0.005 ^c^	0.253 ± 0.002 ^d^	0.289 ± 0.004 ^c^	0.343 ± 0.008 ^d^
16	Isoamyl acetate	0.011 ± 0.002 ^b^	0.897 ± 0.026 ^b^	3.125 ± 0.002 ^b^	2.908 ± 0.001 ^a^	3.280 ± 0.005 ^b^	4.905 ± 0.006 ^a^
17	Ethyl valerate	ND	0.298 ± 0.011 ^b^	0.689 ± 0.005 ^a^	0.607 ± 0.006 ^c^	0.698 ± 0.002 ^c^	0.832 ± 0.003 ^a^
18	Ethyl hexanoate	0.020 ± 0.004 ^a^	1.381 ± 0.004 ^a^	6.459 ± 0.005 ^a^	6.120 ± 0.004 ^a^	6.990 ± 0.008 ^a^	8.908 ± 0.007 ^a^
19	Ethyl lactate	ND	0.674 ± 0.009 ^b^	3.278 ± 0.003 ^b^	3.099 ± 0.008 ^b^	3.201 ± 0.005 ^b^	3.864 ± 0.004 ^a^
20	Ethyl heptanoate	ND	0.050 ± 0.004 ^c^	0.243 ± 0.002 ^a^	0.189 ± 0.002 ^a^	0.278 ± 0.001 ^a^	0.305 ± 0.000 ^c^
21	Ethyl caprylate	0.011 ± 0.002 ^b^	0.572 ± 0.009 ^b^	2.680 ± 0.007 ^a^	2.534 ± 0.002 ^b^	2.723 ± 0.001 ^a^	3.024 ± 0.002 ^a^
22	Isobutyl hexanoate	ND	0.034 ± 0.004 ^d^	0.119 ± 0.008 ^c^	0.107 ± 0.004 ^d^	0.121 ± 0.002 ^d^	0.134 ± 0.002 ^d^
23	Isobutyl lactate	ND	0.063 ± 0.008 ^d^	0.339 ± 0.006 ^a^	0.307 ± 0.000 ^d^	0.329 ± 0.003 ^d^	0.378 ± 0.002 ^b^
24	Ethyl benzoate	ND	0.058 ± 0.007 ^d^	0.302 ± 0.002 ^b^	0.289 ± 0.005 ^d^	0.313 ± 0.000 ^d^	0.367 ± 0.001 ^c^
25	Diethyl succinate	ND	0.108 ± 0.016 ^d^	0.502 ± 0.006 ^b^	0.478 ± 0.004 ^d^	0.539 ± 0.003 ^b^	0.656 ± 0.004 ^a^
26	Ethyl phenylacetate	ND	ND	0.052 ± 0.002 ^c^	0.049 ± 0.003 ^d^	0.053 ± 0.000 ^d^	0.063 ± 0.003 ^d^
27	Ethyl tetradecanoate	ND	0.010 ± 0.003 ^d^	0.061 ± 0.003 ^b^	0.058 ± 0.006 ^d^	0.067 ± 0.002 ^d^	0.082 ± 0.004 ^d^
	Subtotal	0.065	8.800	34.285	33.095	36.505	42.452
	Aldehydes (7)						
28	Acetaldehyde	ND	0.132 ± 0.009 ^c^	0.067 ± 0.005 ^c^	0.056 ± 0.004 ^d^	0.142 ± 0.002 ^a^	0.063 ± 0.003 ^a^
29	1,1-Diethoxyethane	ND	1.001 ± 0.008 ^b^	0.531 ± 0.005 ^b^	0.521 ± 0.003 ^d^	1.031 ± 0.003 ^a^	0.420 ± 0.001 ^a^
30	2-Methylpropionaldehyde	ND	0.209 ± 0.010 ^a^	0.098 ± 0.003 ^a^	0.109 ± 0.003 ^a^	0.199 ± 0.003 ^a^	0.047 ± 0.005 ^a^
31	Acrolein	ND	0.209 ± 0.001 ^a^	0.168 ± 0.004 ^c^	0.165 ± 0.006 ^d^	0.278 ± 0.001 ^b^	0.152 ± 0.001 ^a^
32	Hexanal	ND	0.301 ± 0.012 ^c^	0.175 ± 0.008 ^b^	0.179 ± 0.006 ^d^	0.367 ± 0.004 ^a^	0.167 ± 0.002 ^a^
33	3-Methylbutyraldehyde	ND	0.013 ± 0.001 ^d^	0.050 ± 0.007 ^d^	0.047 ± 0.011 ^d^	0.060 ± 0.000 ^d^	0.052 ± 0.002 ^d^
34	Octanal	ND	0.090 ± 0.006 ^c^	0.061 ± 0.003 ^c^	0.065 ± 0.005 ^d^	0.125 ± 0.002 ^c^	0.057 ± 0.002 ^a^
	Subtotal	ND	0.954	0.619	0.621	1.171	0.538
	Ketones (4)						
35	2,3-Butanedione	0.012 ± 0.003 ^d^	0.132 ± 0.014 ^d^	0.052 ± 0.004 ^d^	0.064 ± 0.004 ^d^	0.148 ± 0.003 ^b^	0.040 ± 0.002 ^a^
36	2-Pentanone	0.034 ± 0.004 ^a^	0.209 ± 0.010 ^a^	0.148 ± 0.003 ^a^	0.136 ± 0.002 ^a^	0.249 ± 0.001 ^a^	0.152 ± 0.004 ^a^
37	2-Heptanone	0.022 ± 0.004 ^c^	0.200 ± 0.013 ^c^	0.136 ± 0.003 ^d^	0.141 ± 0.005 ^b^	0.217 ± 0.002 ^c^	0.108 ± 0.004 ^b^
38	Acetophenone	0.016 ± 0.001 ^b^	0.238 ± 0.004 ^b^	0.103 ± 0.010 ^c^	0.121 ± 0.006 ^d^	0.266 ± 0.002 ^c^	0.104 ± 0.005 ^b^
	Subtotal	0.084	0.779	0.439	0.462	0.880	0.404
	Acids (3)						
39	Acetic acid	0.007 ± 0.001 ^a^	0.010 ± 0.002 ^a^	0.016 ± 0.002 ^a^	0.016 ± 0.002 ^a^	0.018 ± 0.001 ^a^	0.023 ± 0.003 ^a^
40	Hexanoic acid	0.012 ± 0.001 ^a^	0.023 ± 0.002 ^a^	0.031 ± 0.002 ^a^	0.034 ± 0.004 ^a^	0.047 ± 0.000 ^a^	0.062 ± 0.002 ^a^
41	Octanoic acid	0.014 ± 0.002 ^a^	0.025 ± 0.002 ^a^	0.034 ± 0.003 ^a^	0.032 ± 0.002 ^a^	0.041 ± 0.001 ^a^	0.056 ± 0.004 ^a^
	Subtotal	0.033	0.058	0.081	0.082	0.106	0.141
	Others (4)						
42	2-Butylfuran	ND	0.091 ± 0.006 ^c^	0.112 ± 0.005 ^d^	0.107 ± 0.010 ^d^	0.116 ± 0.002 ^d^	0.126 ± 0.002 ^d^
43	Furfural	0.012 ± 0.004 ^c^	0.278 ± 0.023 ^c^	0.123 ± 0.005 ^d^	0.126 ± 0.006 ^d^	0.301 ± 0.001 ^d^	0.203 ± 0.002 ^a^
44	Benzaldehyde	0.011 ± 0.005 ^a^	0.034 ± 0.004 ^a^	0.070 ± 0.005 ^a^	0.065 ± 0.009 ^a^	0.076 ± 0.002 ^a^	0.085 ± 0.006 ^a^
45	Dimethyl trisulfide	ND	0.127 ± 0.002 ^b^	0.264 ± 0.003 ^a^	0.253 ± 0.008 ^d^	0.311 ± 0.001 ^d^	0.324 ± 0.001 ^c^
	Subtotal	0.023	0.530	0.569	0.551	0.804	0.738
	Total	5.765	64.843	100.310	106.922	117.061	128.059

^1^ ND, not detected. ^2^ Data presented in concentration format were expressed as means ± standard deviations obtained from triplicate measurements. Values within the same row that bear different superscript letters denote statistically significant differences (*p* < 0.05).

## Data Availability

The data used to support the findings of this study can be made available by the corresponding author upon request.

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
