# Peer review of "Fungal Community Succession and Volatile Compound Changes during Fermentation of Laobaigan Baijiu from Chinese Hengshui Region"

_foods, 2024, doi:10.3390/foods13040569_

Round 1

Reviewer 1 Report

Comments and Suggestions for Authors

Words like Dipei and Daqu should be explained in the text. They are not obvious for readers

The methodology should be supplemented with production parameters and conditions such as temperature, pH, (samples were taken in industrial conditions, but the process conditions determine the quality of the product, so they should be known)

The scope of research is very wide and well described statistically, which is a very positive.

In my opinion, the authors could write a little more discussion of the results – this is just a suggestion.

As a conclusion of the results – the results provide a theoretical basis for improving the quality of Baijiu. This is a very general statement. A specific solution should be proposed, e.g. which micro-organism to promote to increase/decrease the concentration of selected volatile compounds

Reviewer 2 Report

Comments and Suggestions for Authors

Dear Authors, Thank You for such an interesting paper concerning microbes succession during the fermentation of regional alcoholic beverages. This is new and interesting, also because novel methods of strain analysis were employed by You. 
In my opinion, the manuscript needs many, but minor changes. Please find below my detailed comments:

line 16 : I live in Poland; when starting the reading, it is not clear to me what Daqu and Ercha mean; these are kind of regional expressions in technology. I hope that it is explained in the introduction.

line 34-39: what is Daqu, Dipei ? It should be explained. I live in Poland. I'm aware that each technology and region has its own expressions and names, but as the recipients of this manuscript live around the world, it should be somehow more explained for such people. I know vodka or wine technology but Baiju is something regional to me thus I , as a reader, expect that it will be explained by the authors in the introduction without it is  interesting but difficult to understand at the beginning

line 79: in my opinion, the dimensions of the fermenting vessel should be added here since without it, the "10cm from the bottom" means not too much.

line 166: In my opinion, the statistical analysis for GC-MS results should be added here and then- in tables with the results, at least the standard deviation or variance should be calculated and presented

Figure 1 and Figure 3 and Figure 4 : in my opinion, the symbols will be difficult to read from an A4 printed page,if the authors want to enable the readers to print this article with the use of A4 sheets should consider enlarging the pictures with small symbols

Table 2: with any statistical data it is difficult to tell if these results are statistically different or not. Please calculate and add at least the standard deviation to these values

Reviewer 3 Report

Comments and Suggestions for Authors

Baijiu is a clear liquid usually distilled from fermented sorghum, and the qÅ« starter culture  is used to “baijiu” contains significantly higher alcohol content (35–60%). The specific distilled Baijiu is commonly made from pure sorghum and spontaneous fermented using original fermenters. The fermentation process consists of two solid fermentations stages called Dacha and Ercha.

In this manuscript, the authors have assessed the dynamic changes of fungal communities in different fermentation  stage were revealed by HTS technique, while HS-SPME-GC-MS. detected the corresponding volatile flavor compounds.

The Samples were withdrawn from the upper, the middle and the lower of a ferment tank at six specific fermentation periods. The samples collected on same date were thoroughly mixed and designated as D-1, D-2, D-3, E-4, E-5, E-6, respectively, where D and E represent the two specific fermentation periods call Dacha and Ercha, and 1, 2, 3, 4, 5, 6 represents the critical sampling day during the Laobaigan Baijiu fermentation.

The genomic DNA was rapidly extracted. This was followed by PCR amplification and sequencing. The similarities and differences in the constituent structures of different sample species were determine by a sample hierarchical clustering tree.

The diversity of species abundance distributions among samples was quantified by measuring the significant differences with the sample distance heatmap. The unweighted pair group method with arithmetic mean (UPGMA) approach.

The volatile compounds obtained by SPME extraction were assayed by headspace SPME-GC-MS. The volatile flavor compounds were visualized using heat maps  and classified by cluster analysis. PCA analysis allowed heat map visualization analysis,

This study revealed the dynamic changes of fungal communities and volatile flavor during the fermentation of Hengshui Laobaigan Baijiu. Species classification and abundance analysis showed that Saccharomyces, Saccharomycetales unidentified Candida, Mucor, Aspergillus, and Rhizopus were the predominant genera during fermentation process. Heat map and PCA showed that stage E-5 was critical to the formation of unique aroma of Laobaijian Baijiu and the fermentation period could be brought forward to end at stage E-5.

The PLS-based correlation analysis showed that the fungal Saccharomyces, Candida, Pichia and Trametes had important roles in the formation of volatile compounds.

In conclusion, this study provides a good entry point for an in-depth understanding of the fermentation mechanism of white wine, and provides a theoretical basis for the improvement of quality, flavor and industrial production of Baijiu.

In general, this manuscript is well-written and the scientific assessment of the dynamic changes of fungal communities in different fermentation  stages revealed by HTS technique. Whereas the flavor compounds were detected and quantified by while HS-SPME-GC-MS.

Comments on the Quality of English Language

EXCELLENT
